# Real-Life Experience with Pomalidomide plus Low-Dose Dexamethasone in Patients with Relapsed and Refractory Multiple Myeloma: A Retrospective and Prospective Study

**DOI:** 10.3390/medicina57090900

**Published:** 2021-08-28

**Authors:** Maria Livia Del Giudice, Alessandro Gozzetti, Elisabetta Antonioli, Enrico Orciuolo, Francesco Ghio, Sara Ciofini, Veronica Candi, Giulia Fontanelli, Irene Attucci, Giuseppe Formica, Monica Bocchia, Sara Galimberti, Mario Petrini, Gabriele Buda

**Affiliations:** 1Hematology Unit, Department of Clinical And Experimental Medicine, University of Pisa, 56126 Pisa, Italy; mliviadelgiudice@gmail.com (M.L.D.G.); e.orciuolo@alumni.sssup.it (E.O.); francescoghio83@gmail.com (F.G.); sara.galimberti@med.unipi.it (S.G.); mario.petrini@unipi.it (M.P.); 2Hematology Unit, University of Siena, Azienda Ospedaliera Universitaria Senese Siena, 53100 Siena, Italy; alegozzetti@icloud.com (A.G.); ciofini2@student.unisi.it (S.C.); monica.bocchia@unisi.it (M.B.); 3Haematology Unit, Careggi Hospital-University of Florence, Largo Brambilla 3, 50134 Florence, Italy; elisabettaantonioli@libero.it (E.A.); irene.attucci@gmail.com (I.A.); giuseppe.formica@unifi.it (G.F.); 4UOS Ematologia, San Donato Hospital, ASL8, 52100 Arezzo, Italy; candiveronica86@gmail.com; 5UOC Oncologia, UOS Ematologia, Ospedale San Giuseppe, 50053 Empoli, Italy; giulia.fontanelli@uslcentro.toscana.it

**Keywords:** pomalidomide, dexamethasone, multiple myeloma, immunomodulatory therapy, real-world clinical trials

## Abstract

*Background and Objectives***:** The treatment of Myeloma after the second relapse is still challenging. The aim of the study was to investigate the outcomes of the POM-DEX regimen in real clinical practice. *Materials and Methods:* We retrospectively and prospectively analyzed 121 patients with MM treated with POM-DEX in three Italian sites in Tuscany. We assessed the efficacy based on IMWG Uniform Response Criteria in 106 patients who had received at least two courses of the POM-DEX regimen. The median time from diagnosis to use of POM-DEX was 65 months. POM-DEX median use was in the fourth-line therapy. 63.6% were exposed to lenalidomide or thalidomide, 40.5% to bortezomib or carfilzomib or ixazomib, 5.8% to mAbs in the immediately preceding line of therapy. *Results:* ORR was 43.4%. Median PFS and OS were 8.5 and 14 months. Eighty-nine patients received more than two courses: their median PFS and OS were 11 and 16 months. When used as the third line of therapy, median PFS and OS were 9 and 20 months and, when patients received POM-DEX for more than two courses, median PFS and OS were 14.5 and 22.5 months. *Conclusions:* POM-DEX is effective in RRMM, regardless of the latest exposure to IMiDs, PIs, and mAbs in the previous line of therapy.

## 1. Introduction

Multiple myeloma (MM) is a heterogeneous disease with the uncontrolled clonal proliferation of plasma cells, accounting for approximately 10% of all hematologic cancers [1]. In the absence of curative therapy, the aim of the treatment is to improve overall survival. In recent years, high-dose chemotherapy with autologous stem-cell transplantation and the association with the novel agents such as thalidomide, bortezomib, and lenalidomide, led to a significant improvement in prognosis by increasing the response rates and survival parameters in the general population [2,3,4]. However, the prognosis of patients after a second relapse remains poor, and the treatment is still challenging. For a long time, the only possible approaches for these patients have been palliative therapy or inclusion in clinical trials. Immunomodulatory agents, such as pomalidomide, are active in patients with RRMM [5,6,7]. Pomalidomide (POM) is a third-generation immunomodulatory agent, and its structure is similar to thalidomide and lenalidomide. It has an antiproliferative and proapoptotic action on tumor cells: in fact, it has an immunomodulating action on T and NK lymphocytes, increasing their tumoricidal activity and on regulatory T lymphocytes, decreasing their activity. Moreover, due to its antiangiogenic and anti-inflammatory action, pomalidomide allows reprogramming of the microenvironment [5,6,7,8]. When used in combination with dexamethasone (DEX), pomalidomide exhibits synergistic effects [9]. Thanks to the phase three study MM-003, pomalidomide in combination with dexamethasone (DEX) was approved as a subsequent line of therapy to the second one by the US Food and Drug Administration and the European Medicines Agency (EMA) in February and August 2013, respectively, showing efficacy in patients with RRMM and previously exposed to both bortezomib and lenalidomide, which represent the backbone of the first conventional treatment lines of MM patients [6]. Therefore, starting from these assumptions, we collected real-world evidence data on patients with RRMM who used POM-DEX starting from the third line of treatment.

## 2. Methods

Patients with a previous multi-treated RRMM, who had shown a relapse, disease refractoriness, or drug intolerance during the previous line of therapy, treated with pomalidomide between 2014 and 2021 in three hospital centers in Tuscany (the University hospitals of Florence, Pisa, and Siena) were analyzed retrospectively and prospectively. The study was conducted according to the guidelines of the Declaration of Helsinki and approved by the Ethics Committee. All patients provided written informed consent. Patients were treated with POM following the approved regimen, which is composed as a monthly program, as follows: 4 mg/day (recommended dose), once daily, orally, on days 1 to 21 of repeated courses of 28 days; it should be taken in addition to dexamethasone, 40 mg/day, orally, once a day, in days 1, 8, 15 and 22. Patients aged 75 years old or more received 20 mg. Dose modifications for DEX were in accordance with the institutional guidelines [10]. Every patient received thromboprophylaxis, adjusted for individual risk; myeloid and erythroid growth factors, anti-infectious prophylaxis, treatments for adverse events, and other supportive therapies, including bisphosphonates [11], were based on physicians’ decisions. Treatment was permanently discontinued on withdrawal of the patient’s consent, disease progression, or the occurrence of unacceptable adverse effects. POM treatment was temporarily discontinued or the dose reduced based on the tolerability and safety profile. Prior to receiving pomalidomide treatment, background information of the patients was collected: age, gender, performance status according to Eastern Cooperative Oncology Group (ECOG), and presence of underlying diseases. Baseline laboratory evaluations and clinical features at the time of diagnosis were also analyzed. Cytogenetic analyses were performed using conventional cytogenetics protocols and interphase fluorescence in situ hybridization (FISH). The FISH panel for MM included tests for Immunoglobulin heavy chain (14q32) break apart, translocation of chromosomes 4 and 14, or 14 and 16, and also translocation of chromosomes 14 and 11, 13q14 deletion, 17p13 deletion, and ploidy status. According to the International Myeloma Working Group (IMWG) 2014 Consensus Criteria, we considered cytogenetically detected 17p deletion and translocation of chromosomes 14 and 16 or chromosomes 4 and 14 to indicate a high-risk patient [12]. Response and progression were assessed according to the International Myeloma Working Group consensus criteria [13]. With respect to the last line of treatment before POM-DEX, the relapse was defined as a disease progression in patients who reached at least partial response in the previous treatment; the refractoriness was defined as a disease progression without ever achieving a measurable disease response during the previous treatment or within 60 days after the treatment. The severity of adverse events was evaluated according to version 4.0 of the National Cancer Institute’s (NCI) Common Terminology Criteria for Adverse Events. When two or more adverse events on the same toxicity profile occurred in the same patient, we described only the most serious one of them.

**Statistical analysis** Categorical data were described by absolute and relative frequency, continuous data by median and range. Overall survival (OS) was defined as the time interval from therapy initiation to death. Progression-free survival (PFS) was defined as the time interval from therapy initiation to observed disease progression, relapse, or death from any cause.

## 3. Results

**Patient characteristics** The analysis includes a total of 121 patients with RRMM, treated with POM-DEX in the lines of therapy subsequent to the second (third to seventh) and with median use in the fourth line. Table 1 shows the patients’ baseline characteristics. The median age of all patients at the time of diagnosis was 65 years old; 66 patients were 65 or older than 65 years old, and about one-fifth had International Staging System (ISS) stage III myeloma. Furthermore, 56 patients had received a previous autologous stem cell transplant (single or double). We included patients with performance status (PS) from zero to three points according to ECOG and classified patients’ impairment according to PS: absent (PS 0, 35.5% of the patients), minor (PS 1, 47.1%), moderate (PS 2, 12.4%) or severe (PS 3, 1.7%). Karyotype analysis with interphase FISH was performed in 55 patients; among them, only 9 were found with impaired high-risk cytogenetics (8 with chromosome 14 translocations and 2 patients with 17p13 deletion; 1 patient had both), preventing the analysis of the subgroup. On data cut off (1 February 2021), median survival from initial diagnosis was 84 months. All patients had received at least two previous lines of therapy and, as per guideline, had been exposed both to lenalidomide and bortezomib. As therapy immediately preceded pomalidomide, 75 patients received lenalidomide; 31 received bortezomib; one patient received both lenalidomide and bortezomib; 7 received monoclonal antibodies (mAbs), such as daratumumab (6) and elotuzumab (1); some patients received the newest proteasome inhibitors (PIs), such as carfilzomib (17) or ixazomib (1); 2 patients of the total 77 who received an IMiDs, received thalidomide. Table 2 shows details about immediately prior therapy choices. Every patient received at least one course of POM-DEX; among them, POM-DEX was administered for at least two courses in 106 patients and for at least three courses in 89 patients. POM-DEX was used as the third line of therapy in 55 patients; among them, it was administered for at least two courses in 47 patients and for at least three courses in 36. Sixty-six patients used POM-DEX from the fourth to the seventh line of therapy, and 26 of them received POM-DEX for at least two courses. Ninety-three patients received pomalidomide at the recommended dosage; in 28 patients (23%) a dose reduction was required (to 3 mg in 11 patients and to 2 mg in 17 patients, also commercially available). Based on the immediately preceding line of therapy (mainly lenalidomide, 69.1%), whenever possible, patients were considered relapsed (58) or refractory (57); the group of lenalidomide-based regimens previous line therapy was equally divided between relapsed (36) and refractory (35) patients. Response assessment was performed after a minimum of two cycles of treatment. On the other hand, safety and tolerability were evaluated on a total of 121 patients.

**Efficacy** In a total of 106 patients, the treatment response rate (ORR), including all patients with a partial response or better, was 43.4%. A total of 10 patients gained a very good partial response (8) or a complete response (2). More details regarding the responses to treatment are shown in Table 3. Among relapsed myeloma, 28 gained PR or better, against 15 in the refractory group. ORR was 46.8% in the subgroup of patients who used POM-DEX in the third line and 40.6% in the group of patients who used the regimen from the fourth to the seventh line of therapy. Median progression-free survival (PFS) and overall survival (OS) were 8.5 and 14 months, respectively. In the subgroup who received at least three courses, median PFS and OS were 11 and 16 months, respectively. In the third line setting, median PFS was 9 months, and median OS was 20 months; in patients treated with POM-DEX in third-line therapy for at least three courses, median PFS and OS were 14.5 and 22.5 months, respectively. In patients who had received a previous autologous stem cell transplant, median PFS was 11 months and median OS was 15 months. Based on the immediately preceding line of therapy, patients were divided between relapsed and refractory myeloma: in the former, median PFS was 8 months, and median OS was 17 months; in the latter, median PFS and OS were 6 and 11 months, respectively. In the group of lenalidomide-based regimens, previous line therapy median PFS and OS were 7 and 14 months, respectively, regardless of relapsed or refractory myeloma status. In the general group of 121 patients, analyzed by intention to treat (ITT), median PFS and OS were 6 and 12 months, respectively (Figure 1), with 7 months median PFS and 12 months median OS in the third-line setting.

**Adverse Events** Among 121 patients, 65 experienced one or more clinically relevant adverse events. The most common adverse events were hematologic toxic effects, as shown in Table 4, such as neutropenia (31 patients), anemia (16), thrombocytopenia (3); we also described gastrointestinal symptoms such as diarrhea, infections or sepsis, pneumonia, and other conditions (for example pyrexia, fatigue or rash). Grade three or grade four adverse events occurred in 27 patients and were also mostly hematological ones, such as neutropenia (10 patients), cytopenia (2), and anemia (1) or not otherwise specified (1), followed by pneumonia (2 patients); sepsis occurred in 2 patients due to Pseudomonas aeruginosa and Enterococcus faecalis; cutaneous toxicity (1 patient); 1 patient had grade 4 diarrhea associated with Clostridioides difficile infection. Among grade three and grade four events, we also described acute myocardial infarction, atrial fibrillation, development of a secondary neoplasm (a myelodysplastic syndrome), and a worsening of visual acuity that occurred in 1 patient each. Two patients experienced deep vein thrombosis, and one patient experienced a transient ischemic attack. Thirty-two patients died during the follow-up. The causes of death were mainly myeloma disease progression; one patient died from spontaneous brain hemorrhage unconnected to myeloma.

## 4. Discussion

Multiple myeloma is a malignant hematological disease. Great benefits in terms of survival and improvement in the quality of life were described in recent years thanks to new treatments. Therefore, there is still no definitive algorithm to guide salvage therapy after the second line. However, since clinical trials include highly selected populations, data from real-world evidence may have important practical implications for physicians. In this study, we analyzed the efficacy of oral pomalidomide plus dexamethasone regimen in the population that received more than one cycle of POM-DEX therapy. Although our patients received POM-DEX at an advanced stage of disease (they had a median of three prior lines of therapy and the median time from diagnosis to use of POM-DEX was 65 months), with half of the population of our study refractory to the previous line used and more than half with 65 years old and older, the findings from our real-life experience show that pomalidomide plus low-dose dexamethasone is effective and well-tolerated when sequenced immediately after treatment failure of prior regimens. Our study supports the results obtained experimentally in previous studies, comparing favorably with those from the MM-002, MM-003 (Nimbus), and MM-010 (Stratus) registration studies, which investigated the use of POM-DEX in advanced MM. The ORR reported in our study was 43.4% and is better than those studies (33% in MM-002, 31% in Nimbus, and 32.6% in Stratus). The PFS observed in our case series of 8.5 months is also favorably comparable with that of previously mentioned trials (which described median results of 4.0–4.6 months). These results remain valid even when the sample is analyzed by intention-to-treat. Furthermore, these data show that, while there is a survival benefit associated with the early use of POM-DEX (in the third line of treatment), the benefit of the therapy is not lost even when used in a highly unfavorable prognostic context, i.e., when the patient has had more than three recurrences. In addition, the third-line therapy subgroup ORR was higher than the population that used POM-DEX as subsequent line therapy, and it may also have contributed to the prolonged median PFS.

Moreover, in the population with relapsed disease, survival benefits were greater than in the refractory population. However, in the subpopulation treated with an immediately preceding lenalidomide-based regimen, median OS and PFS are quite comparable with those of the general population despite the fact that in this group, there is an almost equal number of patients considered relapsed or refractory to such treatment. The described results encourage the use of pomalidomide despite previous use of other immunomodulators and support the ability of pomalidomide to induce myeloma response despite close time exposure to lenalidomide, with respect to which it differs not only pharmacokinetically (unlike lenalidomide, of which about 82% is excreted as the parent drug in the urine, 2% of pomalidomide is excreted unchanged through the kidneys and so it can be used with reasonable confidence even in patients with impaired renal function) [14], but also pharmacodynamically [15]. As for the population that had previously received autologous hematopoietic stem cell transplantation (single or double), they experienced better results in terms of survival than the general population. This can be explained not only by recognizing that autologous stem cell transplantation still has a crucial role in the front-line treatment of patients with multiple myeloma but also because transplant candidates typically have better biological baseline characteristics (lower age, fewer comorbidities, adequate response to induction therapies).

In addition, although triplet regimens are widely considered the standard of care in myeloma [16,17], it should be emphasized that the efficacy of POM-DEX, which is a doublet regimen, should not be underestimated for all those patients in which three-drug regimens are not indicated (because they are frail or very elderly, or with significant adverse effects related to proteasome inhibitors, such as peripheral neuropathy) or for whom the use of triplets is hindered by other factors (suburban living, inadequate family support, reluctance to frequent hospital accesses). In the landscape of myeloma treatments, currently polarized on triplet regimens, this study wants to make its contribution to support the use of this regimen for all cases in which the use of multi-drug regimens does not appear to be indicated. The heterogeneous population of patients with RRMM is growing numerically, thanks to the increased life expectancy in the disease, the later age of relapses after the second, due, for example, to the possibility of electing to transplant an age group (those over 65 years old) previously categorically excluded, as well as the introduction of novel agents. There is a need for effective and safe therapeutic options for all those patients for whom three-drug regimens or more aggressive chemotherapies would not be tolerated. About the POM-DEX regimen, considering the very low cost of management, the lack of hospitalization required to administer the drug, the use of fully oral therapy, the possibility to reduce the dose due to the availability of different dosages on the market, and the low impact on the quality of life of patients already treated with multiple drugs, it should be considered as a beneficial treatment option for this heavily pretreated patient population. In addition, it seems significant to us that the best results were achieved in patients who used pomalidomide earlier (in the third line) than in those who used it in the lines subsequent to the third. This suggests that early drug usage is associated with the best desirable outcomes and supports, by extension, the early use of POM even in multi-drug regimens when used in patients who can tolerate them (e.g. in combination with bortezomib, as encouraged by results from the phase 3 OPTIMISM study) [18]. Real-world studies such as ours support the choice of the clinician, who often faces a very heterogeneous, complex, and hard-to-manage population, of which the studies available in the literature are often unrepresentative, and, in this regard, our case series is to our knowledge one of the largest on the use of POMA-DEX in a context of real clinical practice. A point in favor of our study is that our sample also included exposure to recently introduced drugs, not only lenalidomide and bortezomib, as in most modern clinical studies, but also carfilzomib and ixazomib (14.9% of the population), with 5.8% of the population already exposed to monoclonal antibodies (daratumumab and elotuzumab). This study has potential limitations that warrant consideration, including the retrospective design and the lack of a control arm to confirm the efficacy and the safety of the regimen. In addition, further prospective studies on a larger population would improve the quality of the investigation. Given a relatively small sample size and that no formal statistical tests were performed, these results should be interpreted with caution.

## 5. Conclusions

The results of this retrospective study show the high efficacy of the pomalidomide plus dexamethasone regimen as advanced line therapy in RRMM, to be considered where multi-drug regimens cannot be administered, proving the current validity of the results obtained in the previous clinical trials when applied to real clinical practice.

## Figures and Tables

**Figure 1 medicina-57-00900-f001:**
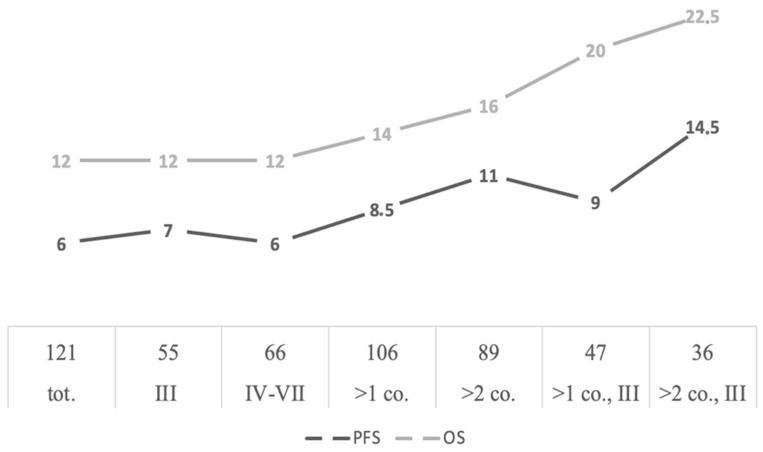
Summary of median OS and PFS analysis. III and IV-VII indicate the line of therapy in which POM-DEX was used. PFS, progression-free survival; OS, overall survival. Tot., total; co., POM-DEX course/courses.

**Table 1 medicina-57-00900-t001:** Patients’ baseline characteristics ^†^.

Age—yr (range) <65 yr 65–87 yr 75–87 yr	65 (38–87) 55 (45.5) 66 (54.5) 16 (13.2)
Gender Male Female	57 (47.1) 64 (52.9)
ECOG 0 1 2 3 ne	43 (35.5) 57 (47.1) 15 (12.4) 2 (1.7) 4 (3.3)
Type of myeloma IgG *κ*	45 (37.2)
IgG *λ*	24 (19.8)
IgA *κ*	9 (7.5)
IgA *λ*	14 (11.6)
Light chain *κ*	13 (10.7)
Light chain *λ*	13 (10.7)
NS/SP	3 (2.5)
ISS stage I II III ne	40 (33) 27 (22.3) 25 (20.7) 29 (24)
Cytogenetic High risk Standard risk	55 (45.5) 9 (7.5) 46 (38)
Previous lines th. II III IV V VI	55 (45.5) 36 (29.7) 22 (18.2) 5 (4.1) 3 (2.5)
Transplantation Autologous (single or tandem) Allogeneic	56 (46.3) 2 (1.7)
Refractory myeloma Relapsed myeloma na	57 (47.1) 58 (47.9) 6 (5)

^†^ Data are median (range) or number (%). Yr, years; ISS, International Staging System; NS, non-secreting (2 patients), SP, solitary plasmacytoma (1 patient); ne, not evaluated; na, not applicable; th., therapy.

**Table 2 medicina-57-00900-t002:** Previous-line treatments.

Regimen ^†^	Total Patients
BVD	10
CVD	1
Ctx	4
Dara	1
DaraRD	2
DaraVD	3
DPACE	2
EloRD	1
IxaRD	1
KD	6
KCD	1
KRD	9
KTD	1
RD	60
RMD	1
TD	1
VD	7
VDMy	6
VDPACE	2
VMP	1
VRD	1

^†^ BVD, bendamustine plus bortezomib-dexamethasone; CVD, cyclophosphamide plus bortezomib-dexamethasone; Ctx, cyclophosphamide; Dara, daratumumab; DaraRd, daratumumab plus lenalidomide-dexamethasone; DaraVD, daratumumab plus bortezomib-dexamethasone; DPACE, dexamethasone plus cisplatin plus doxorubicin plus cyclophosphamide and etoposide; EloRD, elotuzumab plus lenalidomide-dexamethasone; IxaRD, ixazomib plus lenalidomide-dexamethasone; KD, carfilzomib-dexamethasone; KCD, carfilzomib plus cyclophosphamide-dexamethasone; KRD, carfilzomib plus lenalidomide-dexamethasone; KTD, carfilzomib plus thalidomide-dexamethasone; RD, lenalidomide plus dexamethasone; RMD, lenalidomide plus melphalan-dexamethasone; TD, thalidomide plus dexamethasone; VD, bortezomib plus dexamethasone; VDMy, cyclophosphamide plus bortezomib plus liposomal doxorubicin; VDPACE, bortezomib plus doxorubicin plus cisplatin plus cyclophosphamide plus etoposide-dexamethasone; VMP, bortezomib plus melphalan-prednisone; VRD, bortezomib plus lenalidomide-dexamethasone.

**Table 3 medicina-57-00900-t003:** Best response to pomalidomide plus dexamethasone treatment.

Category ^†^	Total Patients 106
CR VGPR PR MD SD PD ne	2 (1.9) 8 (7.5) 36 (34) 9 (8.5) 19 (17.9) 25 (23.6) 7 (6.6)
ORR (≥PR) CRR (sCR/CR) ORR, III ORR, IV-VII	46 (43.4) 2 (1.9) 22 (46.8) 24 (40.6)

^†^ Category of response on the basis of International Uniform Response Criteria. CR, complete response; VGPR, very good partial response; PR, partial response; MD, minimal response; SD, stable disease; PD, progressive disease; ne, not evaluated; ORR, overall response rate; CRR, complete response rate; sCR, stringent complete response. III and IV-VII indicate the line of therapy in which POM-DEX was used.

**Table 4 medicina-57-00900-t004:** Summary of the most commonly reported adverse events.

Neutropenia	31
Anemia	16
Worsening renal function	5
Haematological toxicity nos	3
Pneumonia	4
Skin changes	3
Diarrhea	3
Worsening of cardiac function	3
Leukopenia	3
Thrombocytopenia	3
Dizziness	2
Deep vein thrombosis	2
Pyrexia	1
Sepsis	2
Trilinear cytopenia	2
Atrial fibrillation	1
Decreased appetite	1
Fatigue	1
Muscle spasms	1
Myocardial infarction	1
Secondary neoplasm	1
Worsening of vision	1
Transient ischemic attack	1

## Data Availability

The data presented in this study are available on request from the corresponding author. The data are not publicly available due to privacy and ethical restrictions.

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
