# Peer review of "Real-Life Experience with Pomalidomide plus Low-Dose Dexamethasone in Patients with Relapsed and Refractory Multiple Myeloma: A Retrospective and Prospective Study"

_medicina, 2021, doi:10.3390/medicina57090900_

Round 1

Reviewer 1 Report

In this retrospective study Del Giudice et al reproduce data from other similar real world studies in a cohort of advanced relapsed myeloma patients. Novelty is missing.

Author Response

Reviewer 1

Dear Reviewer, 

Thank you for your feedback. We are aware that this study relies on data that have already emerged in the recent literature. However, we believe that the heterogeneity of previous treatments, the better outcomes than pivotal studies and the stratification that emerged from the number of cycles and from the already experienced therapy lines is relevant in the management of patients with RRMM in the setting of common clinical practice, especially for patients already exposed to the newest agents currently available for Multiple Myeloma.

Reviewer 2 Report

 Authors assessed the efficacy of the POM-DEX regimen in 121 patients with relapsed/refractory multiple myeloma not treatable with triplet regimens and exposed to most new pharmacologic agents, including a subpopulation already exposed also to monoclonal antibodies.

 This is real-would evidence for better assess the therapeutic options for RRMM patients, who represent a population that is both growing and under-represented in clinical trials.

Author Response

Reviewer 2 

Dear Reviewer, 

Thank you for your comment.  We are glad that you have identified as a strength of our study our main goal of providing outcomes in terms of treatment options to a subpopulation with RRMM, already exposed to many novel agents and not eligible for multi-drugs regimens:  this subpopulation is growing in number and is hard-to-manage in common clinical practice!

Reviewer 3 Report

This is a very interesting study, describing real-life outcomes of administering pomalidomide and low dose dexamethasone in relapsed refractory multiple myeloma. The following are relatively minor points regarding presentation and/or clarification, which should be addressed:

  • Whilst the overall presentation is good, there are some very long paragraphs which appear as walls of words. Shortening paragraphs, for example in Methods and Results, will make the script more accessible to the reader.
  • Throughout there are typos Examples include: Line 23 in the Abstract 'in in' should be 'in'. Line 291 should read 'one patient died from spontaneous brain haemorrhage unconnected to myeloma'. A careful proof read should help eliminate these minor aberrations.
  • During the proof reading also check abbreviations. eg ECOG should be annotated in line 114 when Eastern Cooperative Oncology group is first used, rather than line 156.
  • I found it quite difficult to understand the Results section of the Abstract. Can the authors please check that?
  • Line 187/188: 'a dose reduction was required (up to 3mg in 11 patients and up to 2mg in 17 patients'). I'm assuming the dose was reduced TO 3mg or 2mg. Is that correct? Otherwise, the original wording would imply that there was a reduction of 3mg or 2 mg ie from 4mg to 1mg or 2mg.
  • Line 190: does that mean the patients were aged 58years and 57years, or were they patients numbered 58 and 57 in a total of 121? If the former, please insert 'and they were 58 and 57 years old, respectively'.
  • Lines 191-192: 36+35 patients adds up to 71, not 75 patients.
  • Table 1: check numbers throughout. eg assuming 121 patients in study, numbers don't always tally (ages 61+66+16=143); relapsed and refractory myeloma (57+58=115).

Other than these somewhat pedantic points, which still need correction or clarification, the Discussion was good, in particular the information on the effect of previous ASCT, albeit with the acknowledgement that those patients were younger and had fewer co-morbidities. It would have been interesting to see actual outcomes data for ASCT and no ASCT patients. Also interesting was the absence of strong association with acquired cytogenetic aberrations associated with poorer outcomes in these RRMM patients. I note there was no mention of 1q or 1p probes in the iFISH profiles and wonder if these were used. It would have been useful to have information on the iFISH probes used although I acknowledge there may have been differences between the participating centres. I would suggest that the omission of a means, for example, to identify 1q21 amplification, or other aberrations which are mainly seen in later disease, is something that should be rectified in ongoing studies if not already included. Also, in what is obviously a strong regional group, it is hoped that GEP and other molecular tools will become available for these important clinical studies.

Overall then, this is an important study describing real life outcomes of POM-DEX use in RRMM and would be an important addition to the limited literature currently available. 

Author Response

Reviewer 3

Dear Review, 

after re-reading our work in the light of your considerations, we agree with you.

First of all, we have lightened the methods and results sections and updated the results section in the abstract, to make the study easier to read and to understand.

We have corrected the typos in the abstract and made a correction in the cause of death of one of our patients that will make the text easier to understand. 

We have also changed the abbreviation of ECOG, as you correctly pointed out, and clarified the reported dosage of pomalidomide taken for the necessary reductions during treatment. 

With reference to refractory or relapsed patients (former line 190), 57 (in number) belonged to the first category and 58 to the second, as you can assume from Table 1. 

With reference to Table 1, moreover, we have noticed, thanks to you, how some general sets were not easily understandable; we have updated the reported data and we have immediately added the section na (not applicable) in refractory or relapse setting category, under your suggestion, hoping that now the consultation of the Table is more usable and easier for the readers. 

With reference to the former lines 191-192, 71 refers to patients treated with lenalidomide (out of a total of 75) that can be defined as relapsed or refractory; for 4 patients out of 75 treated with lenalidomide, the data allowing their placement in one or the other classification were missing, or the treatment was interrupted due to intolerance; they are included in the so-called not applicable section, i.e. na, of Table 1.

Moreover, we take up your suggestion about the FISH analysis of chromosome 1 aberrations. Unfortunately, for this type of study, which took place over a long period of observation, we did not have the opportunity to test cytogenetics and FISH for a significant number of patients with the probe that you indicate. We reserve (and hope) to be able to report data on chromosome 1 aberrations in future studies, thanks to the fact that our laboratories are becoming equipped with this capability in routine cytogenetics as well as for gene expression profiling data, which we think will soon become increasingly relevant in the daily management of our patients.

We hope you can understand our point of view as authors, and we hope you can appreciate the corrections made to the resubmitted manuscript. We thank you for your contribution and for your point of view.

Round 2

Reviewer 1 Report

There is nothing wrong with this paper but still i cannot find its novelty. Patients that received more than 2 cycles are a selected population of MM patients responding better to POMDEX and prior therapies with new agents are reported in a small subgroup of patients ( 7 Dara 17 Carfilzomib).